# Reactive Molecules in Cigarette Smoke: Rethinking Cancer Therapy

**DOI:** 10.3390/biotech14030052

**Published:** 2025-06-27

**Authors:** Vehary Sakanyan

**Affiliations:** Faculty of Biology, Faculty of Pharmacy, Nantes University, 44035 Nantes, France; vehary.sakanyan@univ-nantes.fr or vehary.sakanyan@orange.fr; Tel.: +33-(0)632623998

**Keywords:** reactive oxygen species (ROS), electrophilic molecules, cancer progression, in-gel detection, regular cigarette, smoking cessation

## Abstract

Science has made significant progress in detecting reactive oxygen species (ROS) in tobacco smoke, which is an important step for precision cancer therapy. An important advance is also the understanding that superoxide can be produced by electrophilic molecules. The dual action of hydrogen peroxide, directly or via electrophilic molecules, in the development of oxidative stress allows for the identification of target proteins that can potentially stop unwanted signals in cancer development. However, despite advances in proteomics, reliable inhibitors to stop ROS-associated cancer progression have not yet been proposed for the treatment of tobacco cigarette smokers. This is likely due to an imperfect understanding of the diversity of molecular mechanisms of anti-ROS action. Fluorescent protein detection in living cells, called in-gel, offers a direct route to a better understanding of the rapid interaction of ROS and electrophilic compounds with targeted proteins. It seemed that the traditional paradigm of pharmaceutical innovation “one drug, one disease” did not solve the problem of tobacco smoking causing cancer. However, among the various therapeutic treatments for tobacco smokers, the best way to combat cancer today is smoking cessation, which fits into the “one-cure” paradigm.

## 1. Introduction

Humanity is facing increasing problems caused by technological progress, which makes people worry about their future. One of the problems affects human health in the fight against cancer, in which cigarette smoking plays a special role. The history of smoking shows unsuccessful attempts to reduce the undesirable effects of tobacco smoking.

Tobacco smoke contains thousands of chemicals, including about 90 chemicals known to cause cancer, which are called carcinogens. Exogenous reactive chemicals can disrupt cellular homeostasis and are often associated with the development of cancer. It is estimated that 90% of human cancers are caused by chemicals, of which 30% are caused by the use of tobacco products, and the rest are caused by chemicals associated with nutrition, lifestyle, and the environment [1]. Of the vast number of agents, more than 100 have been classified as human carcinogens using various methodologies of the International Agency for Research on Cancer (IARC) and the National Toxicology Program’s Report on Carcinogens (RoC).

The tradition of smoking tobacco increases the risk of developing many diseases, including cancer, diabetes, heart attack, stroke, peripheral vascular disease, osteoporosis, chronic obstructive pulmonary disease, and other pathological processes [2]. Among these diseases, cancer is of great importance, which occurs in various organs and tissues and accounts for about 20% of cancer cases diagnosed in the United States due to tobacco smoking [3]. In other countries, the smoking rate is also very high, amounting to every fifth person or even more.

What molecules in tobacco smoke trigger the chemical and biological processes that lead to cancer?

Life arose from the movement of electrons about 2.4 million years ago on Earth [4]. Early removal of reactive molecules was essential for life, so ROS removal mechanisms were evolutionarily advantageous for future generations of living organisms. The field of free radical biology emerged with the discovery of superoxide dismutase (SOD) in 1969. Reactive oxygen species trigger a cascade of events, including DNA mutation and oxidative DNA damage, which are key events in the onset of carcinogenesis [4]. An imbalance between ROS generation and detoxification of radical species causes oxidative stress in cells and tissues, in which H_2_O_2_ plays a role as a mediator in signaling pathways [5].

Electrophilic molecules have a positive charge and a deficiency of electron pairs, allowing them to react with electron-rich atoms in nucleophiles, sharing electron pairs (Figure 1) [6]. Electrophiles form covalent bonds with proteins, nucleic acids, and lipids leading to toxicity. Electrophilic substitution reactions are chemical reactions in which an electrophile replaces a group in a compound. Chemical carcinogens are highly reactive electrophiles, which have electron-deficient atoms that can react with nucleophilic, electron-rich sites in the cell [7].

Biologically active electrophiles are formed in the cell through controlled metabolic processes or as oxidative by-products in unregulated pathological processes. Electrophilic molecules influence various signaling pathways, modulating cytoplasmic kinase and phosphatase or nuclear transcription factor activity in inflammatory diseases [8]. Regular cigarettes (called combustible cigarettes) have tobacco, added chemicals, filter, and a paper covering. When smoking regular cigarettes, the human body is exposed to thousands of chemicals.

New methods based on high-throughput genome sequencing and highly sensitive molecular analysis of genes, RNA, proteins, and lipids, as well as large comparative studies involving patients and healthy individuals, have demonstrated that the processes leading to carcinogenesis in humans are more complex than previously thought, as summarized by Hanahan D. and Weinberg R.A. over the past 25 years [9,10,11].

This review substantiates the role of reactive oxygen molecules and reactive electrophiles in tobacco smoke that lead to cancer development.

## 2. Results

### 2.1. Nicotine and Tobacco Smoking

Of the approximately 60 species of tobacco plants in the genus Nicotiana, only two species, *N. rustica* and *N. tabacum*, are used to make cigarettes [12]. Tobacco smoke is classified as a human carcinogen, and the history of cigarette smoking provides an example for understanding the development of cancer. Tobacco products are addictive because they contain nicotine. Scientific research into the effects of nicotine on biology and behavior is extensive. The nicotine molecule is a tertiary amine consisting of a pyridine moiety and exists in two forms: (S)-nicotine, which binds stereo-selectively to nicotinic cholinergic receptors, and (R)-nicotine, which is present in small amounts in cigarette smoke due to racemization during pyrolysis, is a weak agonist of the same receptor [13]. Nicotine exerts its physiological effects by binding to nicotinic acetylcholine receptors (nAChRs), which are expressed on both neuronal and non-neuronal cells throughout the body [14]. Nicotine has adverse effects on the respiratory, cardiovascular, renal and reproductive systems, suggesting that it is a causative agent of cancer [15].

Recent studies suggest that nicotine is a potential carcinogen that may promote cancer growth and progression through epigenetic effects [16,17]. Tobacco cigarette smoking is a risk factor for non-small cell lung cancer (NSCLC), in which nicotine may play a role in promoting cancer progression through epigenetic regulation of target proteins [17].

However, most scientific studies have failed to provide reliable evidence of the involvement of nicotine contained in traditional tobacco cigarettes in the pathogenesis of cancer. Nicotine makes it more likely that people will continue to use tobacco products even if they want to quit smoking regular cigarettes.

### 2.2. Cancer Development and the Consequences of Smoking

Chemotherapy was first used in clinical practice in 1940 using nitrogen mustard to treat cancer patients [18]. Advances in chemical synthesis have led to the development of targeted therapy drugs that block the growth and spread of cancer cells without affecting healthy cells [18,19]. Exogenous factors such as ionizing radiation, photochemical reactions, environmental toxins, and endogenous biochemical and enzymatic processes within the cell induce the formation of free radicals [20]. The process of radical scavenging results in the replacement of the missing electron that damages DNA, RNA, proteins, or lipids. Therefore, free-radical-generating proteins have become potential therapeutic targets to eliminate disease-associated microenvironmental crosstalk, leading to extensive testing of the antioxidant capacity of proteins.

The human genome encodes 538 kinases, making these enzymes an attractive model for assessing the effects of potential therapeutic molecules, of which only a few molecules have been approved for clinical use [21]. The oxidative stress theory proposes that reactive molecular species underlie disease states, and in vitro and in vivo models have been used to demonstrate their role in the onset and progression of human diseases. Given the limited success of potential drugs in counteracting the negative effects of reactive molecules, the clinical reliability of protein targeting in cancer therapy often remains questionable. An analysis of the treatment effect showed that non-smokers appeared to have longer overall survival and progression-free survival compared with former or current smokers [22]. However, this analysis was based on indirect comparisons, and more robust trials with direct comparisons were considered necessary to draw definitive conclusions.

Smoking status does not affect the clinical outcomes based on immune checkpoint inhibition in patients with metastatic renal cell carcinoma, as observed in a clinical trial [23]. More recently, the impact of tobacco smoking on survival outcomes in patients receiving targeted therapy has also been studied in renal cell carcinoma on a larger scale [24]. The results of these studies clearly showed that patients who smoke have significantly worse survival outcomes than patients who have never smoked or who have quit smoking.

Tobacco smoke contains a mixture of chemicals that act as oxidants in redox reactions and can damage cellular targets. The chemical structure of oxygen with two unpaired or single electrons in the outer orbits is quickly transformed into aggressive forms, free radicals, which react with various chemical structures in living organisms. Given the high oxidation-reduction potential of dioxygen (O_2_), it is an ideal electron acceptor, which leads to the formation of reactive oxygen species (ROS), playing an important role in redox reactions at various biological processes [25,26].

Reactive oxygen species are highly reactive free radicals, including superoxide radical (O_2_•^−^), hydrogen peroxide (H_2_O_2_), and hydroxyl radical (•OH) (Figure 2). ROS can be converted into each other by reducing O_2_ to superoxide radical (O_2_•^−^). Excessive ROS production causes oxidative damage to a wide range of biomolecules, including nucleic acids, proteins, carbohydrates, and lipids, which can lead to impaired cellular function and the development of oxidative stress in the human body. The increase in the number of reactive oxygen species in cancer is associated either with an increase in their production by exogenous agents or with the suppression of antioxidant action [27,28,29].

The toxicological effects of smoking are explained by a number of mechanisms. Various observations support the key role of smoking-induced ROS in oxidative stress during inflammation and carcinogenesis. In particular, hydroxyl radicals generated by aqueous cigarette tar can cause oxidative DNA damage [30]. Cigarette smoke and inhaled fibers act synergistically to increase the production of harmful hydroxyl radicals. Cigarette filters with antioxidant compounds impregnated with activated carbon may have only a minor effect on the composition and toxicity of the solid and gaseous phases of cigarette smoke [30]. ROS, especially H_2_O_2_, play an important role in promoting both cell proliferation and tumor cell survival by triggering redox-signaling cascades [31]. Cigarette-smoke-induced oxidative stress is widely recognized as one of the key molecular events mediating the pathogenesis of smoking-associated diseases [32,33].

The best-known mechanism by which the H_2_O_2_ molecule acts as a mediator of cellular signaling is the reversible oxidation of a specific cysteine residue in redox-sensitive proteins that perform regulatory functions in cellular metabolism [34]. The main protein targets involved in redox regulation are tyrosine phosphatases, which, together with tyrosine kinases, maintain a homeostatic status of cysteine phosphorylation to regulate signaling events [35].

### 2.3. Antioxidants

In a short time, original approaches to changing the redox state of tumor cells have been developed, including selective inhibition of ROS sources, hyperactivation and modulation of antioxidant enzymes to maintain healthy ROS levels and thereby further stimulate apoptosis [36]. An antioxidant is a relatively stable, low-molecular-weight compound that can donate an electron to a free radical and neutralize it, thereby reducing its drift potential. Antioxidants act through a variety of mechanisms as radical scavengers, hydrogen donors, electron donors, peroxide scavengers, singlet oxygen quenchers, enzyme inhibitors, synergists, or metal chelators. In addition to natural sources of antioxidants, it has been shown that chemically synthesized agents can also provide similar activity [37].

Organisms living under aerobic conditions are subject to oxidative stress due to damage to cellular macromolecules by reactive molecules formed during cellular respiration or inflammation. In DNA, the guanine base is the richest in electrons and most prone to oxidation, leading to the formation of several oxidation products, the main one being 8-oxo-7,8-dihydroguanine. When DNA is damaged by free radicals, the amount of 8-oxo-7,8-dihydroguanine increases, allowing this molecule to be used as a marker of oxidative stress [38]. Oxidative damage to proteins affects the activity of enzymes and receptors, leading to increased susceptibility to proteolysis and disruption of signaling pathways [39]. Endogenous reactive oxygen and nitrogen species, reactive carbonyls and other electrophiles formed during inflammation may be the main cause of chemical damage to DNA, proteins, and lipids [40]. Chemical antioxidants have been tested to slow the progression of cancer, neurodegenerative diseases, cardiovascular diseases, and other pathologies.

Biological thiols such as glutathione and N-acetylcysteine play an important role in maintaining redox homeostasis by acting as antioxidants and free radical scavengers [41]. The tripeptide antioxidant glutathione, composed of glutamic acid, cysteine, and glycine, is part of the endogenous defense against ROS. N-acetylcysteine provides the cysteine in the glutathione structure and improves the intracellular -SH content and hence the glutathione pool. In healthy cells and tissues, more than 90% of the total glutathione level is in the reduced thiol form, and the remainder is in the oxidized disulfide form. In global cellular redox homeostasis, disulfide–glutathione is maintained at a low level, while reduced thiol–glutathione remains high [42,43]. Therefore, an elevated glutamine disulfide/thiol ratio indicates oxidative stress.

A number of compounds in plants have been found to be natural antioxidants because they have the ability to control free radical formation, stop free-radical-mediated chain reactions, suppress oxidative stress, enhance endogenous antioxidant defenses, and exhibit strong free-radical-scavenging properties, as well as exert positive effects on the pathogenesis of cancer, diabetes, neurodegenerative diseases, and aging [44,45].

Other natural antioxidants such as bee products honey, propolis, and royal jelly act as radical scavengers [45,46]. Evaluation of the antioxidant activity of bee products is necessary to select the best samples that meet the requirements of medical use. Otherwise, a decrease in the redox-buffering capacity in the patient’s body, leading to a violation of the antioxidant–oxidant balance, can cause an inflammatory response, thereby contributing to the progression of cancer and other diseases [41,47].

### 2.4. ROS in Cancer Development and Progression

Reactive oxygen species trigger a cascade of events, including DNA mutation and oxidative DNA damage, which are key events in the onset of carcinogenesis. An imbalance between ROS generation and detoxification of radical species causes oxidative stress in cells and tissues, in which H_2_O_2_ plays a role as a mediator in signaling pathways.

ROS signals are transduced to effector molecules, and this process is regulated by converting non-selective ROS reactions into stable and controlled electrophilic signaling mediated by 8-nitroguanosine 3′,5′-cyclic monophosphate [48]. This suggests that ROS functions are regulated by endogenous electrophiles, which are themselves generated from ROS during various physiological and pathophysiological reactions in the cell. Stress perception and cellular defense activity are regulated by three activated transcription factors, NRF2 (Nuclear factor E2-Related Factor 2), HIF1 (Hypoxia-Inducible Factor 1), and HSF1 (Heat Shock Factor 1), which, respectively, control the antioxidant response, the response to hypoxia, and the removal of unfolded proteins in response to heat shock [49]. That is, the NRF2 protein acts as a master regulator of cytoprotective processes [50,51], playing an important role in maintaining cellular redox homeostasis and regulating the antioxidants glutathione and thioredoxin, as well as stimulating the expression of enzymes involved in the reduction in reactive oxygen species [52,53]. Superoxide dismutase (SOD) catalyzes the reduction in superoxide anions to hydrogen peroxide, and together with catalase and glutathione peroxidase, the activity of these enzymes is manifested by dimeric and higher oligomeric forms [54]. There are three superoxidases in the human body: SOD1 located in the cytoplasm, SOD2 in the cell membrane, and SOD3 outside the cell, which affect about two thousand proteins and cause oxidative changes [55,56]. It is pertinent to note that the Nrf2 defense system requires more than 30 min for protein synthesis and following binding to proteins [57,58].

The oligomerization state of ROS molecules was studied using electron microscopy and protein profiling analysis in breast and prostate cancer cells exposed to the electrophilic compound nitrobenzoxydiazole (NBD) [59]. A significant proportion of Cu/Zn-SOD1 was rapidly transformed into a 32 kDa dimer in cells exposed to lipophilic compounds, whereas the activity forms of two other enzymes, catalase and glutathione peroxidase, remained monomeric.

Furthermore, a method for directly detecting the binding of small fluorescent dyes to proteins immobilized on a nitrocellulose membrane (in-gel binding assay) demonstrated rapid binding (within 5 min) of the dye to fifteen highly expressed proteins in breast cancer cells, and the total number of moderately and weakly expressed targets attacked by H_2_O_2_ exceeded hundreds of target proteins [60,61]. This important observation suggests that the global ROS defense system lags in its ability to protect damaged proteins from electrophilic attack in cancer cells due to the longer time required for transcription and translation of the NRF2 protein.

Because the electrophilic NBD structure binds to a large number of different proteins via H_2_O_2_ generation, this scenario suggests more complex processes in which electrophilic molecules and reactive molecular species can act in concert on target proteins. The functional action of electrophilic molecules mediated by superoxide generation resembles that of cytoplasmic SOD1, which may explain the importance of other enzymes in initiating oxidative stress in the cell. The hypothetical concept of ROS generation according to the classical scenario may involve not only the SOD1 enzyme but also other electrophilic substitutions in targeted proteins, as shown in Figure 3.

Historically, long-term attempts to use chemically synthesized inhibitors of the epidermal growth factor receptor (EGFR), including fourth-generation drugs, in the fight against cancer have not been successful [29]. The emergence of resistant mutants in EGFR is considered to be the reason for the ineffectiveness of drugs, some of which have even been officially approved for cancer treatment. However, there is another explanation. The ineffectiveness of EGFR inhibitors is due to the attempt to remove from the cell an enzyme that plays an important role in many biochemical and biological functions, to which cancer cells respond by creating drug-resistant mutants [29].

Thus, the rapid degradation of many target proteins by NBD molecules excludes a rescue role for the NRF2 system, which requires at least thirty minutes. These data are important for better understanding the harmful effects of tobacco cigarette smoking on human health, which are clearly visible in patients, but the true cause and mechanism of action remain unknown.

In summary, two types of reactive species, H_2_O_2_ and NBD, target multiple proteins in living cells, and the dual action of reactive species exacerbates the deleterious effects, ultimately leading to cell death. The severity of oxidative stress primarily depends on the rate of SOD1 dysfunction and presumably on other targeted proteins. Given that the nucleophilicity of amino acids in the cell depends on various factors, including the redox status, it is conceivable that the number of bound proteins, including non-specific interactions, may be different under different culture conditions. Even at low doses, electrophilic species can irreversibly bind to proteins and thus unpredictably contribute to the deregulation of cellular metabolism. Therefore, identification of proteins covalently reacting with NBD compounds in living cells will help to establish a list of suitable diagnostic and therapeutic biomarkers of electrophilic stress as a prerequisite for the development of new strategies to combat cancer.

### 2.5. On the Way to Quitting Smoking

The level of chemical impact on the human body increases negatively when smoking regular tobacco cigarettes. Such negative impact is associated with the formation of many toxic forms of electrophilic and reactive molecules in the lungs of smokers at high temperatures of about 1000°. To reduce the harmful effects of tobacco, modified forms of cigarettes have been developed in which filters have been changed, or the composition of tobacco has been changed in order to reduce the formation of harmful molecules at high smoking temperatures. Cigarette manufacturers emphasize in their advertising the absence of harmful effects of cigarette smoke and the safety of cigarette smoking. However, experimental assessment shows only a slight decrease in reactive forms of molecules in cigarette smoke but in a significant amount to cause a pathological effect on the body. In addition, at present there are no large-scale clinical data allowing us to talk about a new generation of “harmless” cigarettes.

Lung cancer in never-smokers is a common cause of cancer death, and whole-genome sequencing of 232 never-smokers identified three subtypes defined by copy number aberrations [62]. Although no clear evidence of tobacco smoking was found, genetic modifications in the receptor tyrosine kinase-RAS pathway have a clear impact on survival, as five genomic alterations independently doubled mortality in never smokers. Tumor sections collected at surgery and follow-ups in 421 patients with NSCLC were also analyzed. Despite a history of smoking, 8% of lung adenocarcinomas showed no evidence of tobacco-induced mutagenesis. But these tumors had similar rates of EGFR mutations and oncogenic isoforms of RET (rearranged during transfection), ROS1 (reactive oxygen species proto-oncogene 1), and ALK (anaplastic lymphoma kinase positive), compared to tumors in never smokers, suggesting that they have similar etiology and pathogenesis [63].

Epidemiological studies have shown that smokers suffer more lung cancer and other respiratory diseases than non-smokers. When smoke passes through a burning cigarette filter, reactive oxygen species and other harmful molecules enter the lungs and cause damage to the respiratory system. The free-radical-scavenging effect can be assessed by comparing tobacco smoke from a filter with and without an antioxidant for comparison, as shown in Figure 4.

Natural antioxidants such as green tea extract, tomato extract, and grape seed extract have been incorporated into rod filters to eliminate toxic radicals and reduce airway damage. The most attractive result was obtained with nano-cerium, a rare earth nanomaterial with catalytic activity that mimics several types of enzymes by transferring electrons between their oxidation states. Nano-cerium functionalized with alendronate shows good scavenging capacity for reactive radicals, similar to SOD enzymes [64]. Another nanozyme, biochar isolated from silkworm excrement, demonstrated antioxidant properties that could scavenge excess free radicals and protect lung tissue in mice exposed to tobacco smoke [65].

The tobacco industry’s desire to maintain a very large market has led to the development of a new type of cigarette called “Heated Tobacco Products” (HTPs) or “Heat-not-Burn” (HnB), advertised as harmless. However, like all other tobacco products, they are toxic in nature and contain carcinogens. HTP and HnB cigarettes produce aerosols containing nicotine and toxic chemicals that are inhaled by smokers when they smoke.

Our team found that grape melanin is a fairly strong antioxidant and was tested to assess the ability to scavenge free radicals from tobacco smoke of improved and supposedly safer cigarettes offered by several companies. A decrease in the optical density of a solution containing the sensitive reagent 2,2-diphenyl-1-picrylhydrazyl was shown [66]. The burning rate of cigarettes was monitored by residual pressure using a manometer. The data obtained (Table 1) show that the use of grape melanin in the iQOS (“I Quit Original Smoking”) cigarette filter leads to a decrease in optical density by about 56% compared to the optical density of smoke in an iQOS cigarette treated with water. The temperature reduction from 884 °C on the filter of a regular tobacco cigarette to 250–350 °C in an iQOS cigarette shows that the biological protection of the grape melanin provides a significant reduction in the amount of harmful chemicals during smoking. These results indicate that during smoking of HTPs cigarettes, the grape melanin is indeed able to reduce the number of active reactive molecules. However, the effect of removing chemical compounds is insufficient and does not provide grounds for talking about complete purification of HTPs cigarettes from harmful molecules.

What chemicals in tobacco smoke can negatively affect the body when smoking? From over 4000 smoke constituents, 69 have been considered as possible or proven carcinogens [67] (Table 2). Benzofurans, including 4-nitro-2,1,3-benzoxydiazole, are found in high concentrations in unfiltered cigarettes (200–300 μg/cigarette), which is high enough to be a potential carcinogen among the major components of particulate matter in unfiltered cigarette smoke. Most of the carcinogenic compounds found in cigarette smoke are not present in the native tobacco leaf but are formed by pyrolysis at high temperatures during cigarette combustion.

Smoking is associated with chronic and progressive inflammation, which is a key factor in the pathophysiological development of tobacco-related diseases [68]. The inflammatory microenvironment is characterized by elevated levels of reactive molecular species, leading to oxidative stress. The link between cancer occurrence and risk factors is that almost 50% of cancer deaths worldwide are caused by smoking and alcohol consumption. Breast, tracheal, bronchial, and lung cancers are the most significant cancer burden globally [69]. More men die from cancer than women, as men tend to smoke and drink more alcohol than women. Significant progress has been made in reducing tobacco exposure, which can be attributed to coordinated international and national cancer prevention efforts ROS formation can be stimulated by various exogenous agents, including pollutants, dietary agents, drugs, lifestyle factors, or radiation. In this regard, smoking continues to be the leading risk factor for cancer worldwide, as tobacco smoke contains thousands of chemicals, including potential carcinogens [70].

There are effective smoking cessation treatments that can be integrated into clinical oncology care in a variety of ways. Meta-analyses of 400 studies conducted up to 2012 in patients with lung or head and neck cancer showed improved survival with smoking cessation among these patients [71,72]. Survival analyses among 4526 smokers (2254 women) with cancer treated between 2006 and 2022 confirmed that smoking cessation treatment within 6 months to 5 years after cancer diagnosis maximizes survival benefits. These data support smoking cessation as an important early clinical intervention for patients after cancer diagnosis [73]. Early screening and diagnosis of lung cancer is therefore particularly important, which may not only improve the prognosis of lung cancer through early medical intervention but may also encourage diagnosed patients to change their habitual risk behavior.

Since 2017, the National Cancer Institute (USA) has been investing heavily in the Smoking Cessation Initiative to expand access to smoking cessation treatment for cancer patients [74]. This example has inspired other organizations around the world to join the initiative. It is hard to disagree with the statement that smoking patients should recognize that among cancer treatment options, smoking cessation is one of the most effective approaches in terms of improving survival, quality of life, and overall health. Reducing exposure to harmful risk factors associated with cigarette smoking may not only have a positive impact on efforts to reduce the cancer burden but also synergistically improve population health.

Electronic cigarettes, also known as e-cigarettes or vapes, are portable electronic devices that mimic the effects of a tobacco cigarette by producing vapor when inhaled instead of smoke [75,76,77]. The advantage of vapes over tobacco cigarettes is that the difference lies in the composition of the smoking material, and tobacco is burned at a lower temperature, about 300 °C. Because vapes do not contain tobacco, they are a different type of smoking product than regular tobacco cigarettes.

Smoking e-cigarettes is a relatively new phenomenon, and there are no large-scale medical studies to determine their safety. Therefore, it is unclear whether vaping is safer than smoking tobacco, as advertised. Large-scale epidemiological studies are needed to assess the association between long-term e-cigarette use and the risk of lung cancer.

## 3. Conclusions

Smokers constitute a significant portion of the world’s population, and epidemiological studies have shown that they suffer more from lung cancer and other respiratory diseases than non-smokers. Various types of chemical antioxidants have been developed to treat oxidative stress in cancer in smokers, but none have demonstrated real therapeutic efficacy.

A number of plant antioxidants, such as extracts from tea, cocoa, and other vegetables and fruits that produce sulforaphane, resveratrol, silymarin, alpha-tocopherol, and curcumin, are undergoing clinical trials for the treatment and prevention of various diseases, including cancer [78,79,80,81]. However, overall clinical data on the therapeutic use of antioxidant plant products remain limited and sometimes contradictory [82]. Therefore, further research is needed to fully understand their mechanisms of action, optimal dosage, and possible medical effects and implications.

The synthesis of electrophilic chemical protein inhibitors, which can be considered a failure in cancer treatment, has advanced human thought in another direction—the chemical synthesis of compounds that destroy targeted cancer proteins. How much more effective this new approach will be on a global scale compared to the synthesis of protein inhibitors is still unclear. Given that our planet is exposed to ROS and other reactive molecular species through two mechanisms, namely, firstly, natural oxidants (reactive molecule formation on a global scale, volcanoes, and industrial emissions) and, secondly, synthesized electrophilic chemicals, targeting the degradation of the responsible proteins seems an attractive but apparently limited way to treat patients. Therefore, the reduction in industrial emissions should be considered in the context of protecting human health in the natural environment.

On the other hand, knowledge about the action of natural and chemically synthesized anti-ROS compounds has advanced in recent years but does not yet allow us to propose a definitive therapeutic approach and a reliable cure for cancer. It is possible that further analysis of a large number of molecules with known structures can advance our understanding of the specific mechanism of action of selected compounds in the fight against cancer in non-smokers and cancer progression in smoking patients. The results of AI-based analysis can be simplified by integration with registry systems when assessing the activity of chemical molecules as possible therapeutic candidates for various types of cancer. In any case, a real understanding of the action of carcinogens among many thousands of chemical agents gives hope for solving an important medical problem.

Currently, given the lack of a real therapeutic barrier to the development and progression of smoking-induced cancer, quitting smoking conventional and modified cigarettes is the best way to prevent cancer.

## Figures and Tables

**Figure 1 biotech-14-00052-f001:**
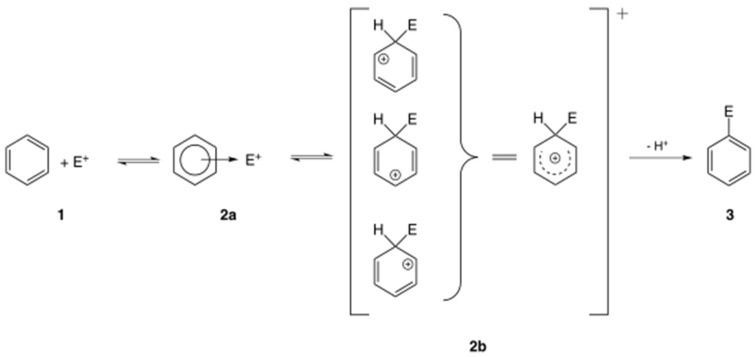
General mechanism of electrophilic compound substitution [6].

**Figure 2 biotech-14-00052-f002:**
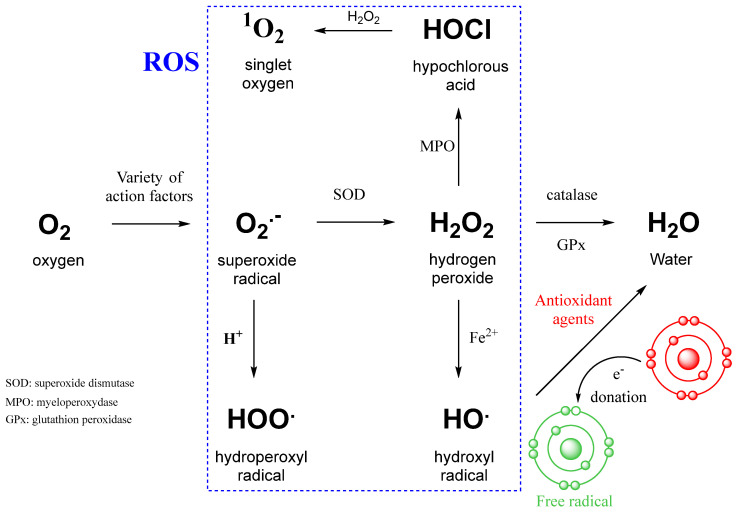
Intracellular conversion between different types of ROS and the action of antioxidant agents [29]. SOD—superoxide dismutase; MPO—myeloperoxidase; GPx—glutathione peroxidase; Fe^2+^—ferrous iron involved in the Fenton reaction.

**Figure 3 biotech-14-00052-f003:**
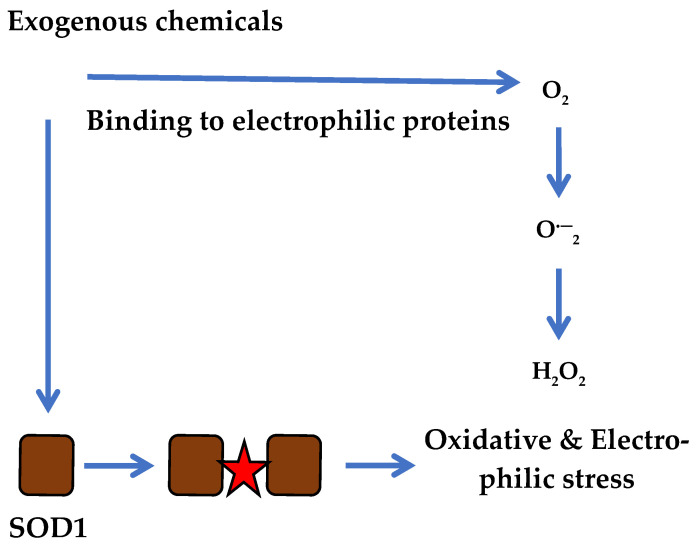
Progression of electrophilic and oxidative stress in cancer cells simultaneously via SOD1 and multiple electrophilic reactions induced by reactive compounds.

**Figure 4 biotech-14-00052-f004:**
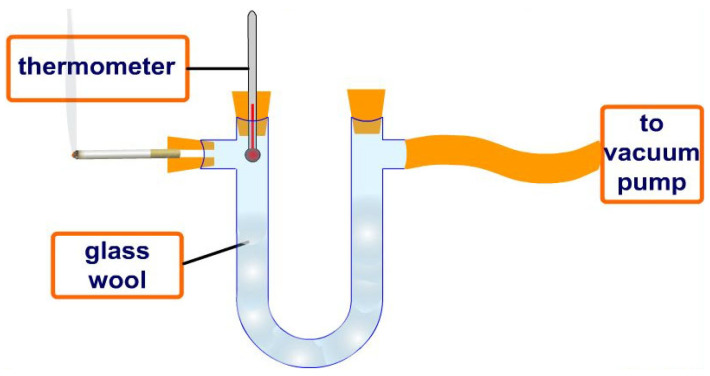
Conventional removal of harmful chemical compounds from smoke of burning tobacco cigarettes using a filter impregnated with a potential antioxidant to evaluate the smoke purification effect.

**Table 1 biotech-14-00052-t001:** Free radical scavenging in the modified iQOS cigarette [66].

Sample	Optical Density of DPPH Solution
Original iqos cigarette	0.379
Filter treated with water	0.355
Filter impregnated with grape melanin	0.199

**Table 2 biotech-14-00052-t002:** Major components of particulate matter in unfiltered cigarette smoke [67]. Only a small number of the thousands of chemicals are shown for comparison with the compound benzofuran.

Compound	µg/Cig.
Nicotine	100–3000
Nornicotine	5–150
Anatabine	5–15
Anabasine	5–12
Total non-volatile HC	300–400
Naphthalenes	3–6
Pyrenes	0.3–0.5
Phenol	80–160
Other Phenols	60–180
Catechol	200–400
Other Catechols	100–200
Other Dihydroxybenzenes	200–400
Palmitic Acid	100–150
Quinones	0.5
Solanesol	600–1000
Linoleic Acid	150–250
Indole	10–15
Quinolines	2–4
Benzofurane	200–300

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
