# Peer review of "Reactive Molecules in Cigarette Smoke: Rethinking Cancer Therapy"

_biotech, 2025, doi:10.3390/biotech14030052_

Round 1

Reviewer 1 Report

Comments and Suggestions for Authors

The work represents an interesting and broad summary for the understanding of a relevant and current topic also supported by many updated bibliographic references. The multidisciplinary approach based on the integration of chemical, molecular and clinical knowledge in a coherent way is also interesting. The discussion of the interaction between ROS and electrophilic molecules is well structured and supported by experimental data. The use of the NBD dye to identify electrophilic stress in cancer cells is an original and promising contribution. The language is generally clear. However, there are some critical points that require improvement:

  • The speech is often long and dispersive with some concepts that are repeated without adding new elements making the reading redundant (e.g. the role of H₂O₂ is treated in several places in a redundant way).
  • To avoid overlapping between the various sections, a clearer and more distinct structure of physio-pathological, molecular, experimental, and clinical information is suggested.
  • It is suggested to review the bibliography as it is not correctly reported in the text (for example, reference 16 should replace reference 14 in the manuscript).
  • Some references appear to be excessively self-referential (multiple publications by the author himself).
  • It is suggested to provide a more extensive explanation of the “in-gel” method for non-expert readers.
  • To clarify the innovative therapeutic proposal with respect to existing data in the literature.

In conclusion, it is suggested that the manuscript be reviewed by the author.

Comments on the Quality of English Language
  • The speech is often long and dispersive with some concepts that are repeated without adding new elements making the reading redundant (e.g. the role of H₂O₂ is treated in several places in a redundant way).

Author Response

To Reviewer 1.

Comments and Suggestions for Authors

The work represents an interesting and broad summary for the understanding of a relevant and current topic also supported by many updated bibliographic references. The multidisciplinary approach based on the integration of chemical, molecular and clinical knowledge in a coherent way is also interesting. The discussion of the interaction between ROS and electrophilic molecules is well structured and supported by experimental data. The use of the NBD dye to identify electrophilic stress in cancer cells is an original and promising contribution. The language is generally clear. However, there are some critical points that require improvement:

  • The speech is often long and dispersive with some concepts that are repeated without adding new elements making the reading redundant (e.g. the role of H₂O₂ is treated in several places in a redundant way).

Response: The text has been shortened where necessary to avoid repetition.

  • To avoid overlapping between the various sections, a clearer and more distinct structure of physio-pathological, molecular, experimental, and clinical information is suggested.

Response: Where necessary, the relevant text has been clarified.

  • It is suggested to review the bibliography as it is not correctly reported in the text (for example, reference 16 should replace reference 14 in the manuscript).

Response: The location of some references has been changed.

  • Some references appear to be excessively self-referential (multiple publications by the author himself).

Response: The presentation of our own results has been revised and shortened.

  • It is suggested to provide a more extensive explanation of the “in-gel” method for non-expert readers.

Response: The relevant passage has been slightly modified. Considering your previous remark, I have not provided the details described in the two references 59 and 60.

  • To clarify the innovative therapeutic proposal with respect to existing data in the literature.

Response: I believe that the Abstract and improved text in two chapters and the Conclusion now better reflect my personal views on the issue of tobacco smoking in the revised manuscript.

In conclusion, it is suggested that the manuscript be reviewed by the author.

            Response: The previous text has been supplemented and revised. New references have been added. I hope the manuscript has been improved.

Comments on the Quality of English Language

  • The speech is often long and dispersive with some concepts that are repeated without adding new elements making the reading redundant (e.g. the role of H₂O₂ is treated in several places in a redundant way).

Response: Necessary corrections have been made.

     New links have been added.

     All questions and comments have been answered.

Dear Reviewer, Thank you very much. Your criticism and suggestions were helpful and allowed me to improve the quality of this review.

Reviewer 2 Report

Comments and Suggestions for Authors

This review article provides a comprehensive and insightful overview of the role of reactive molecules in cigarette smoke and their implications for cancer therapy. The manuscript highlights significant advances in understanding the molecular mechanisms underlying oxidative stress and carcinogenesis related to tobacco smoking. The discussion on the dual action of hydrogen peroxide and electrophilic molecules, as well as the challenges in developing effective therapeutic inhibitors, is particularly valuable. However, certain sections would benefit from further clarification and additional detail. Specific points in the introduction, results, and conclusion require more precise explanations or supporting evidence to strengthen the overall impact of the review.

Questions 

Introduction

  1. The author mentions that 90% of human cancers are caused by chemicals, with 30% attributed to tobacco products. Could the author clarify the sources and methodologies behind these epidemiological estimates to provide stronger support for these statistics?

  2. The introduction discusses the evolutionary advantage of ROS removal mechanisms. Could the author elaborate on how these ancient mechanisms are relevant to current strategies for cancer prevention or therapy in the context of tobacco-induced oxidative stress?

Results

  1. The review states that nicotine does not cause cancer but contributes to continued tobacco use. Could the author provide more detail on the indirect roles of nicotine in cancer progression, particularly through modulation of cellular pathways or enhancement of exposure to other carcinogens?

  2. The manuscript references the limited clinical reliability of protein targeting in cancer therapy due to the complexity of ROS-related mechanisms. Could the author specify which protein targets have shown the most promise or failure in recent clinical trials?

  3. The review discusses the impact of smoking status on survival outcomes in patients receiving targeted therapies. Could the author clarify whether these findings are consistent across different cancer types and therapeutic modalities?

  4. The section on antioxidants describes various mechanisms of action. Could the author provide examples of specific synthetic or natural antioxidants that have demonstrated efficacy in clinical or preclinical models of tobacco-related cancers?

Conclusion

  1. The author concludes that quitting smoking is currently the best way to prevent cancer in tobacco users. Could the author discuss any emerging therapeutic strategies that may provide additional benefit for former smokers who remain at elevated cancer risk?

  2. The review emphasizes the need for better understanding of anti-ROS mechanisms. What are the most critical knowledge gaps that should be addressed in future research to enable the development of effective ROS-targeted therapies for tobacco-related cancers?

Author Response

To Reviewer 2

This review article provides a comprehensive and insightful overview of the role of reactive molecules in cigarette smoke and their implications for cancer therapy. The manuscript highlights significant advances in understanding the molecular mechanisms underlying oxidative stress and carcinogenesis related to tobacco smoking. The discussion on the dual action of hydrogen peroxide and electrophilic molecules, as well as the challenges in developing effective therapeutic inhibitors, is particularly valuable. However, certain sections would benefit from further clarification and additional detail. Specific points in the introduction, results, and conclusion require more precise explanations or supporting evidence to strengthen the overall impact of the review.

Introduction

1. The author mentions that 90% of human cancers are caused by chemicals, with 30% attributed to tobacco products. Could the author clarify the sources and methodologies behind these epidemiological estimates to provide stronger support for these statistics?

Response 1: Additional information to the text (ref. 1).

2. The introduction discusses the evolutionary advantage of ROS removal mechanisms. Could the author elaborate on how these ancient mechanisms are relevant to current strategies for cancer prevention or therapy in the context of tobacco-induced oxidative stress?

Response 2: The introduction contains additional information supported by 4 new references. Additions also to the chapter "Conclusion".

Results

1. The review states that nicotine does not cause cancer but contributes to continued tobacco use. Could the author provide more detail on the indirect roles of nicotine in cancer progression, particularly through modulation of cellular pathways or enhancement of exposure to other carcinogens?

Response 1: Smoking tobacco cigarettes is a risk factor for NSCLC and possibly other cancers. Two recent papers (refs 16,17) explain this by a possible nicotine-based mechanism, namely epigenetic regulation of certain proteins.

2.  The manuscript references the limited clinical reliability of protein targeting in cancer therapy due to the complexity of ROS-related mechanisms. Could the author specify which protein targets have shown the most promise or failure in recent clinical trials?

Response 2: The lack of beneficial effects of SOD1 inhibitors on cancer may be explained by the importance of this enzyme for human cell function. In particular, inhibition of the active site of some target tyrosine kinases, such as EGFR, by the corresponding inhibitors has no real therapeutic effect on cancer therapy. For more details, see also our published review (ref. 29).

3. The review discusses the impact of smoking status on survival outcomes in patients receiving targeted therapies. Could the author clarify whether these findings are consistent across different cancer types and therapeutic modalities?

Response 3: EGFR plays an important role in establishing cellular homeostasis. Many years of attempts to treat cancer with chemically synthesized EGFR inhibitors have not yielded any real positive results in either smokers or non-smokers. A detailed explanation can be found again in ref. 29.

4. The section on antioxidants describes various mechanisms of action. Could the author provide examples of specific synthetic or natural antioxidants that have demonstrated efficacy in clinical or preclinical models of tobacco-related cancers?

Response 4: Chapter 2.3 and the revised Conclusion add further information on plant antioxidants to highlight their role in controlling ROS formation and suppressing oxidative stress. Plant antioxidants may enhance antioxidant defense, as suggested by clinical trials.

Conclusion

1. The author concludes that quitting smoking is currently the best way to prevent cancer in tobacco users. Could the author discuss any emerging therapeutic strategies that may provide additional benefit for former smokers who remain at elevated cancer risk?

Response 1: Years of attempts to offer reliable cancer therapy using chemically synthesized electrophilic EGFR inhibitors and other multifunctional enzymes have failed. The development of targeted protein degraders offers new hope for smokers and nonsmokers alike, as suggested in the extended Conclusion. Another possibility is offered by exploiting nature's ability to produce antioxidant molecules in plants.

The chemical synthesis of targeted protein-degrading compounds against cancer and other diseases is an alternative way to fight cancer. Hopefully, in the near future it will become clear which approach is effective on a global scale.

2. The review emphasizes the need for better understanding of anti-ROS mechanisms. What are the most critical knowledge gaps that should be addressed in future research to enable the development of effective ROS-targeted therapies for tobacco-related cancers?

Response 2: See also Conclusion. Knowledge about the action of natural and chemically synthesized anti-ROS compounds has advanced in recent years, but does not yet allow us to propose a definitive therapeutic approach and a reliable cure for cancer. A number of plant antioxidants have been clinically tested for the treatment and/or prevention of various diseases, including cancer, neurological diseases, diabetes, etc., despite some negative results about their usefulness. New studies demonstrate the potential of natural antioxidants in medical applications. Further studies are needed to fully understand their mechanisms of action, optimal dosage, and potential drug interactions.

Dear Reviewer, I appreciate your attention to my review and your insightful thoughts on the course and treatment of cancer in smoking and non-smoking patients.

Reviewer 3 Report

Comments and Suggestions for Authors

Review.

Journal BioTech (ISSN 2673-6284)

Manuscript ID biotech-3653621

Type Review

Title Reactive molecules in cigarette smoke: Rethinking cancer therapy

Authors Vehary Sakanyan *

Section Medical Biotechnology

The aim of this article is to consider stopping the progression of ROS-associated cancer in cigarette smokers. In this sense, the author hypothesizes that this is probably due to a poor understanding of the diversity of molecular mechanisms of anti-ROS action.

General considerations

1-The article being presented as a literature review deserves an update with the most recent publications concerning this subject.

2- The article is more of a narrative than a literature review. There are long paragraphs concerning generalities on the subject that could be shortened. The article would benefit from being better structured around 1- the definition (specificity of ROS and cancers in relation to smoked tobacco), 2- epidemiology (country,world), 3- pathophysiology of different cancers and ROS  in relation to cigarettes (molecular composition of tobacco, smoke, effect of coupling with high temperature) 4- Abstention, Seuvrage, prevention, therapeutics, monitoring...

Detailed considerations

Line 32 : Phrased in this general way in this first sentence, one gets the impression that "technological progress" is harmful. It is better to modulate this statement.  Line 41 :Add a more recent reference than (1) is desirable as: Chen S, Ding Y. A Bibliometric Analysis on the Risk Factors of Cancer. Genes Chromosomes Cancer. 2025 Jan;64(1):e70019. doi: 10.1002/gcc.70019. PMID: 39835786. 

Lines 50 and 55 : add a more recent reference than (4) 1960 ! as : Guo Q, Tang Y, Wang S, Xia X. Applications and enhancement strategies of ROS-based non-invasive therapies in cancer treatment. Redox Biol. 2025 Mar;80:103515. doi: 10.1016/j.redox.2025.103515. Epub 2025 Jan 28. PMID: 39904189; PMCID: PMC11847112. Or : Xiong B, Zhang Y, Liu S, Liao S, Zhou Z, He Q, Zhou Y. NOX Family: Regulators of Reactive Oxygen Species Balance in Tumor Cells. FASEB J. 2025 Apr 30;39(8):e70565. doi: 10.1096/fj.202500238RRR. PMID: 40266050; PMCID: PMC12017260.

Line 47 : This number must be supported by a bibliographic reference. Lines 68 and 70 : add a recent reference than (7) as (Mucha et all 2021). 

Lines 71-74 : This paragraph requires a transition with the preceding and a bibliographic reference. The transition must also be made with the following for better understanding. 

Line 92 : add ref (Sun and all2024). Line 308 : the abbreviation NSCLC appearing for the first time must be expressed in full.

 Line 311 : the abbreviations  RET, ROS1, ALK, and MET must be expressed in full.

Line 331 : the legend and figure 4 require a reference.

 Line 352 : the legend table 1 require a reference. Line 377 : a reference is needed for table 2.

  Add recent reference…  2021.. 2025… -        Chen S, Ding Y. A Bibliometric Analysis on the Risk Factors of Cancer. Genes Chromosomes Cancer. 2025 Jan;64(1):e70019. doi: 10.1002/gcc.70019. PMID: 39835786.

  • Guo Q, Tang Y, Wang S, Xia X. Applications and enhancement strategies of ROS-based non-invasive therapies in cancer treatment. Redox Biol. 2025 Mar;80:103515. doi: 10.1016/j.redox.2025.103515. Epub 2025 Jan 28. PMID: 39904189; PMCID: PMC11847112.

  • Xiong B, Zhang Y, Liu S, Liao S, Zhou Z, He Q, Zhou Y. NOX Family: Regulators of Reactive Oxygen Species Balance in Tumor Cells. FASEB J. 2025 Apr 30;39(8):e70565. doi: 10.1096/fj.202500238RRR. PMID: 40266050; PMCID: PMC12017260.

  • Mucha P, Skoczyńska A, Małecka M, Hikisz P, Budzisz E. Overview of the Antioxidant and Anti-Inflammatory Activities of Selected Plant Compounds and Their Metal Ions Complexes. Molecules. 2021 Aug 12;26(16):4886. doi: 10.3390/molecules26164886. PMID: 34443474; PMCID: PMC8398118.

  • Sun Q, Jin C. Cell signaling and epigenetic regulation of nicotine-induced carcinogenesis. Environ Pollut. 2024 Mar 15;345:123426. doi: 10.1016/j.envpol.2024.123426. Epub 2024 Jan 29. PMID: 38295934; PMCID: PMC10939829.

Author Response

The aim of this article is to consider stopping the progression of ROS-associated cancer in cigarette smokers. In this sense, the author hypothesizes that this is probably due to a poor understanding of the diversity of molecular mechanisms of anti-ROS action.

General considerations

1-The article being presented as a literature review deserves an update with the most recent publications concerning this subject.

Response: Recent publications have been reviewed for relevance to the review idea and included where appropriate.

2- The article is more of a narrative than a literature review. There are long paragraphs concerning generalities on the subject that could be shortened. The article would benefit from being better structured around 1- the definition (specificity of ROS and cancers in relation to smoked tobacco), 2- epidemiology (country,world), 3- pathophysiology of different cancers and ROS  in relation to cigarettes (molecular composition of tobacco, smoke, effect of coupling with high temperature) 4- Abstention, Seuvrage, prevention, therapeutics, monitoring...

Response: I appreciate your suggestion. But it is preferable to keep the different approach of the review to describe the two global sources of reactive molecules on our planet and the effect of smoking tobacco cigarettes on cancer. I think your suggestions are not ignored, they are just presented differently.

Detailed considerations

Line 32 : Phrased in this general way in this first sentence, one gets the impression that "technological progress" is harmful. It is better to modulate this statement.  Line 41 :Add a more recent reference than (1) is desirable as: Chen S, Ding Y. A Bibliometric Analysis on the Risk Factors of Cancer. Genes Chromosomes Cancer. 2025 Jan;64(1):e70019. doi: 10.1002/gcc.70019. PMID: 39835786. 

Response: The first phrase is slightly modified. I prefer to retain reference 1, an important and first publication on the global assessment of factors leading to the development and progression of cancer, including tobacco cigarette smoking. Your reference is included as n°7.

Lines 50 and 55 : add a more recent reference than (4) 1960 ! as : Guo Q, Tang Y, Wang S, Xia X. Applications and enhancement strategies of ROS-based non-invasive therapies in cancer treatment. Redox Biol. 2025 Mar;80:103515. doi: 10.1016/j.redox.2025.103515. Epub 2025 Jan 28. PMID: 39904189; PMCID: PMC11847112. Or : Xiong B, Zhang Y, Liu S, Liao S, Zhou Z, He Q, Zhou Y. NOX Family: Regulators of Reactive Oxygen Species Balance in Tumor Cells. FASEB J. 2025 Apr 30;39(8):e70565. doi: 10.1096/fj.202500238RRR. PMID: 40266050; PMCID: PMC12017260.

Response: Lines 50-55 are important in terms of the age of our planet.

Added Ref. Go et al.

Line 47 : This number must be supported by a bibliographic reference. Lines 68 and 70 : add a recent reference than (7) as (Mucha et all 2021).

Response: Sorry, I don't agree to replace a great review with this article.

Lines 71-74 : This paragraph requires a transition with the preceding and a bibliographic reference. The transition must also be made with the following for better understanding. 

Response: Transition has been made.

Line 92 : add ref (Sun and all2024). Line 308 : the abbreviation NSCLC appearing for the first time must be expressed in full.

Response: Reference on Sun et al. is included as n°16.

NSCLC means non-small-cells-lung-cancer (corrected).

Line 311 : the abbreviations  RET, ROS1, ALK, and MET must be expressed in full.

Response: Full names have been added to the abbreviations, Lines 367-369.

Line 331 : the legend and figure 4 require a reference.

Response: This drawing was offered on the Internet without attribution or link. I have modified it slightly and believe that attribution and links are not required.

Line 352 : the legend table 1 require a reference. Line 377 : a reference is needed for table 2.

Response: Reference 66 for legend of the table 1 is 57. Reference 67 for Table 2.

  • Guo Q, Tang Y, Wang S, Xia X. Applications and enhancement strategies of ROS-based non-invasive therapies in cancer treatment. Redox Biol. 2025 Mar;80:103515. doi: 10.1016/j.redox.2025.103515. Epub 2025 Jan 28. PMID: 39904189; PMCID: PMC11847112.

Response: To overcome the limitations of ROS-based non-invasive therapy, nanoparticle systems have been proposed to combat relevant cancer types.

Based on your suggestion, I have also added a recent articles that show epigenetic effects of nicotine on cancer development.

Add recent reference…  2021.. 2025… -        Chen S, Ding Y. A Bibliometric Analysis on the Risk Factors of Cancer. Genes Chromosomes Cancer. 2025 Jan;64(1):e70019. doi: 10.1002/gcc.70019. PMID: 39835786.

  • Xiong B, Zhang Y, Liu S, Liao S, Zhou Z, He Q, Zhou Y. NOX Family: Regulators of Reactive Oxygen Species Balance in Tumor Cells. FASEB J. 2025 Apr 30;39(8):e70565. doi: 10.1096/fj.202500238RRR. PMID: 40266050; PMCID: PMC12017260. 
  • Mucha P, Skoczyńska A, Małecka M, Hikisz P, Budzisz E. Overview of the Antioxidant and Anti-Inflammatory Activities of Selected Plant Compounds and Their Metal Ions Complexes. Molecules. 2021 Aug 12;26(16):4886. doi: 10.3390/molecules26164886. PMID: 34443474; PMCID: PMC8398118.

Response: These three suggested sources do not provide new information to improve the quality of the manuscript and are therefore not included in this review.

Dear Reviewer, Thank you very much for your questions and suggestions.

Round 2

Reviewer 2 Report

Comments and Suggestions for Authors

All issues identified in the first review have been properly addressed. The manuscript has been significantly improved and I recommend acceptance.

Author Response

Comments 1: All issues identified in the first review have been properly addressed. The manuscript has been significantly improved and I recommend acceptance.

Response 1: Since there are no further comments from the reviewer, I would just like to express my gratitude to him for reviewing my work.

Reviewer 3 Report

Comments and Suggestions for Authors

It is regrettable that the author of this literature review refuses to update his article by including recent publications such as...

           Chen S, Ding Y. A Bibliometric Analysis on the Risk Factors of Cancer. Genes    Chromosomes Cancer. 2025 Jan;64(1):e70019. doi: 10.1002/gcc.70019. PMID: 39835786.

  • Xiong B, Zhang Y, Liu S, Liao S, Zhou Z, He Q, Zhou Y. NOX Family: Regulators of Reactive Oxygen Species Balance in Tumor Cells. FASEB J. 2025 Apr 30;39(8):e70565. doi: 10.1096/fj.202500238RRR. PMID: 40266050; PMCID: PMC12017260. 
  • Mucha P, Skoczyńska A, Małecka M, Hikisz P, Budzisz E. Overview of the Antioxidant and Anti-Inflammatory Activities of Selected Plant Compounds and Their Metal Ions Complexes. Molecules. 2021 Aug 12;26(16):4886. doi: 10.3390/molecules26164886. PMID: 34443474; PMCID: PMC8398118.

Author Response

Response 1: My selected list of publications on anticancer drugs is quite large, and I had to select the most relevant articles to cite. The manuscript by Mucha R. et al. can be added to the reference list. The other two references are less relevant because they do not contain any new or valuable information about “smoking”.

I hope you agree with my decision. Thank you for the comment.
